# Identification of Key Candidate Genes for Muscle Growth in Liaoning Black Pigs and Duroc Pigs via Longissimus Dorsi Muscle Transcriptome Analysis

**DOI:** 10.3390/cimb47110917

**Published:** 2025-11-05

**Authors:** Zhanpeng Jia, Jiani Li, Fubo Qiao, Jiashuo Zhang, Xianjun Liu, Jing Chen

**Affiliations:** College of Animal Science & Veterinary Medicine, Shenyang Agricultural University, Shenyang 110866, China; m15641247509@163.com (Z.J.); 2023220589@stu.syau.edu.cn (J.L.); fuboqiao@stu.syau.edu.cn (F.Q.); jiashuozhang@stu.syau.edu.cn (J.Z.)

**Keywords:** RNA-seq, pig, longissimus dorsi muscle, muscle growth

## Abstract

Pig growth is an economically important trait regulated by multiple genes and signaling pathways. To explore the molecular mechanisms underlying muscle growth, RNA sequencing was conducted to compare the transcriptomic profiles of the longissimus dorsi muscle between indigenous Liaoning Black pigs (CH) and commercial Duroc pigs (HD). Muscle samples from six CH (three males and three females) and six HD (three males and three females) pigs were analyzed. Functional annotation, Gene Ontology (GO) and KEGG enrichment, and protein–protein interaction (PPI) analyses were performed. Sequencing yielded 12 high-quality datasets (Q20 > 97%, Q30 > 93%). Comparative analysis identified 5051 DEGs in females (CHF vs. HDF; 2310 upregulated and 2681 downregulated) and 9972 DEGs in males (CHM vs. HDM; 4984 upregulated and 4988 downregulated). GO terms were mainly enriched in organonitrogen compound and protein metabolic processes, while KEGG pathways were enriched in focal adhesion and insulin signaling. PPI analysis highlighted hub genes *ITGB1*, *SRC*, *MYL2*, *PRKACA*, and *MAPK3*. qPCR validation showed strong agreement with RNA-seq data. These findings provide valuable insights into the molecular basis of divergent muscle growth between pig breeds.

## 1. Introduction

Pork serves as a primary source of animal protein in the human diet, making research on pork yield and meat quality a major focus in the field of animal husbandry [1]. Research has focused on understanding the regulatory mechanisms behind pig growth and development to increase their growth rates and meat quality. Many factors affect the growth and development of pigs, among which skeletal muscle development is one of the main determinants of production performance, as it directly affects pork yield and quality. Additionally, skeletal muscle serves as a major tissue for both meat production and metabolism [2,3] and possesses a unique cellular architecture. Skeletal muscle fibres are the largest cells in mammals, accounting for 75–90% of skeletal muscle tissue [4]. The number, size, and type of muscle fibres determine the basic characteristics and yield of muscle. Muscle fibre characteristics are closely related to pork color, pH, tenderness, and intramuscular fat (IMF) content [5,6]. IMF content affects muscle tenderness and flavour, with higher IMF content being associated with higher pork quality [7]. Muscle fibre type is a key factor affecting pork quality, and muscle growth after birth mainly depends on the thickening and transformation of muscle fibres [8,9]. These postnatal changes are a continuation of the tightly regulated myogenic program established during early development, in which genetic and transcriptional control determines the number and phenotype of muscle fibres. Skeletal muscle fiber formation and transformation occur in distinct developmental windows—from prenatal primary and secondary fiber formation to postnatal hypertrophy and plastic conversion between oxidative and glycolytic fibers [10,11,12]. Recent multi-omics and histological studies have further shown that this transition remains active during the finishing period (~6–7 months in pigs), particularly between 150 and 210 days of age, when dynamic remodeling of fiber-type composition and transcriptional regulation still occur [13,14,15]. Therefore, this stage represents a critical regulatory window for transcriptomic remodeling and breed-specific muscle growth differences in pigs.

The Liaoning Black pig is an indigenous dual-purpose (meat–fat) breed originating from northeastern China. Its development can be traced to the late 1930s and early 1940s, when Berkshire boars were introduced to improve the genetic composition of local Northeast Chinese pigs. Through successive generations of selective breeding, distinct regional strains such as Dandong Black and Changtu Black were established. Current populations are primarily distributed in Dandong, Benxi, and Changtu of Liaoning Province. The breed is characterized by a medium body size, sturdy constitution, strong adaptability to cold environments, high reproductive capacity, and superior meat quality with moderate intramuscular fat and fine marbling. Genetically, the Liaoning Black pig preserves alleles linked to favorable muscle quality and lipid deposition, making it an important maternal resource for crossbreeding programs. As a result of long-term natural adaptation and artificial selection, the breed has evolved a high tolerance to low temperatures and the capacity to efficiently utilize coarse feed resources typical of northeastern China. The Duroc pig is a US-origin meat-type breed developed in the early nineteenth century from the “Old Duroc” and “Red Jersey” lines. Through intensive selection, modern Durocs exhibit a medium-to-large frame, red coat, drooping ears, robust musculature, and strong feed efficiency; they are widely used as terminal sires in three-breed crossing to enhance growth rate and lean yield [16]. Genomic studies reveal selection signatures and candidate genes associated with fast growth and high lean mass, and document population differences (e.g., American vs. Canadian Duroc) at loci related to growth traits [17]. From a production and meat-quality perspective, Duroc lines show reliable performance for average daily gain, feed efficiency and primal-cut yields, while offering breeding potential for intramuscular fat (IMF) and water-holding/tenderness traits; long-term BLUP-based multi-trait selection has improved IMF without compromising growth [18]. In commercial systems, using Duroc as the terminal sire with Landrace and Large White dams improves growth and carcass traits of crossbred pigs across diverse management conditions [16].

In recent years, transcriptomic approaches have been extensively applied to elucidate the molecular mechanisms governing skeletal muscle development and meat quality in pigs. These studies have revealed numerous genes and signaling pathways involved in muscle fiber differentiation, lipid metabolism, and myofibrillar protein synthesis. Nevertheless, most previous investigations have primarily focused on lean commercial breeds such as Duroc, Landrace, and Large White, whereas comprehensive comparative analyses incorporating indigenous Chinese breeds with distinct meat-quality traits remain limited. Furthermore, the molecular basis underlying breed- and sex-specific transcriptional regulation of muscle fiber composition during the finishing stage has not been fully characterized. Therefore, a comparative transcriptomic investigation of CH and HD is expected to elucidate breed- and sex-specific transcriptional profiles, identify key differentially expressed genes and enriched signaling pathways associated with muscle development and fiber-type composition, and thereby enhance our understanding of the molecular regulatory mechanisms underlying phenotypic variation in muscle growth.

## 2. Materials and Methods

### 2.1. Animals and Sample Collection

Six healthy CH (three males and three females) and six HD (three males and three females) with comparable body weights were selected for sampling. At the time of sampling, the CH were 228 days old with an average body weight of 129.6 ± 2.4 kg, whereas the HD were 202 days old with an average body weight of 123.8 ± 2.1 kg. These two time points were selected to ensure comparable physiological stages and similar final body weights during the finishing phase, allowing the comparison to reflect genetic and transcriptional differences rather than variations in growth rate. All animals were raised at the Liaoning Changtu Liaoning Black Pig Breeding Center under identical feeding and environmental management conditions. The indoor temperature was maintained at 16–22 °C, relative humidity at 60–70%, and a natural 12 h light: 12 h dark photoperiod was used. Pigs were housed on dry, well-ventilated floors, and pens were cleaned and disinfected daily. Animals were fed three times per day with a standard corn–soybean–based diet formulated according to the nutrient requirements of the NRC (2012) for growing–finishing pigs, with fresh water provided ad libitum. The total feeding period lasted no less than 200 days. All experimental pigs were slaughtered on the same day, following the humane slaughtering protocol approved by Shenyang Agricultural University. Muscle samples were collected from the middle portion of the longissimus dorsi, located on the left side of the carcass between the 10th and 12th ribs, to ensure anatomical consistency. All pigs were slaughtered at the slaughtering center affiliated with the Liaoning Changtu Liaoning Black Pig Breeding Base. Approximately 100 mg of tissue was excised within 10 min after exsanguination, immediately frozen in liquid nitrogen on site, and subsequently stored at −80 °C for preservation. Total RNA extraction was performed within two weeks after collection to minimize degradation risk. After RNA extraction, the remaining tissue samples were maintained at −80 °C for subsequent verification or additional analyses.

### 2.2. RNA Extraction and Quality Assessment, Library Construction, and Illumina Sequencing

Total RNA was extracted from the longissimus lumborum muscle using TRIzol^®^ Reagent (Invitrogen, Carlsbad, CA, USA), following the manufacturer’s protocol. The extracted RNA was subsequently treated with RNase-free DNase I (Promega, Madison, WI, USA) to remove residual genomic DNA contamination. The concentration and purity of RNA were evaluated using a BioPhotometer (Eppendorf, Hamburg, Germany), and samples with A260/A280 ratios between 1.9 and 2.1 were considered acceptable. RNA quality and integrity were preliminarily verified via 1% agarose gel electrophoresis and further assessed using the RNA Nano 6000 Assay Kit on the Agilent 2100 Bioanalyzer system (Agilent Technologies, Santa Clara, CA, USA). All RNA samples exhibited RNA Integrity Number (RIN) values greater than 7.0, meeting the quality requirements for transcriptome sequencing. The qualified RNA was stored at −80 °C until use.

Upon passing quality control, the samples were submitted to Beijing Aowei Gene Technology Co., Ltd., (Beijing, China) for library preparation and sequencing. The constructed libraries were sequenced on the Illumina HiSeq 4000 (Illumina, San Diego, CA, USA) high-throughput sequencing platform for transcriptome analysis. Raw reads containing adapters, with undetermined base (N) content exceeding 10%, or with more than 50% of bases having a Phred quality score (Q) ≤ 20 were removed. The filtered clean reads were aligned to the reference transcriptome via TopHat2, and the resulting alignments were assembled via Cufflinks. The consistency of the sequencing results was further assessed through RNA-Seq correlation analysis.

### 2.3. Gene Expression Analysis

The most direct indicator for measuring gene expression levels is transcript abundance: higher transcript abundance corresponds to higher gene expression levels. Since transcript amplification during experiments occurs randomly, transcript abundance is reflected in the sequencing data volume, specifically the number of reads aligned to the corresponding gene. To enable comparability across genes of varying lengths, different experimental conditions, and diverse sequencing depths, the FPKM metric was introduced. Fragments per kilobase of exon model per million mapped reads (FPKM) quantify the number of fragments originating from a specific gene per kilobase of gene length per million mapped reads. By simultaneously accounting for the effects of sequencing depth and gene length on read counts, FPKM has become the most widely adopted method for quantifying gene expression levels [19]. Gene expression analysis was performed for each sample via HTSeq software (version 2.0.5) [20,21] in the union counting mode. Typically, FPKM thresholds of 0.1 or 1.0 are employed to determine gene expression status. In our analytical pipeline, subsequent analyses were restricted to genes with FPKM > 1. Using the read count data derived from the gene expression analysis, differential expression analysis was conducted with DESeq2 [22]. Volcano plots were generated to visualize the overall distribution pattern of the differentially expressed genes. DEGs were identified using the criteria of a q value < 0.05 (Benjamini–Hochberg adjusted *p* value following multiple testing correction) and an absolute |log_2_(FoldChange)| > 1.

### 2.4. Functional Annotation

Gene Ontology (GO; http://www.geneontology.org/ accessed on 20 August 2025) enrichment analysis of differentially expressed genes was implemented using the GOseq R package (version 1.66.0) [23], in which gene length bias was corrected, and all expressed genes detected in the RNA-seq dataset were used as the background gene set. KOBAS (version 3.0) software was used to test the statistical enrichment of DEGs in the KEGG (http://www.genome.jp/kegg/ accessed on 20 August 2025) pathways [24].

### 2.5. Construction of Protein–Protein Interaction Networks for Differentially Expressed Genes

The protein–protein interaction (PPI) network of the DEGs was constructed using the STRING database [25] (v12.0; https://string-db.org accessed on 1 September 2025) with the species parameter set to *Sus scrofa* and a minimum required interaction confidence score of 0.9 to ensure high-confidence interactions. The resulting interaction data were subsequently imported into Cytoscape [26] (v3.10.3) for network visualization and further analysis.

### 2.6. Validation of Differentially Expressed Genes with Quantitative Real-Time PCR (qPCR)

To confirm the repeatability and reproducibility of the RNA-seq gene expression profiles in CH and HD pigs, quantitative real-time PCR (qPCR) was performed on eight randomly selected differentially expressed genes (DEGs) derived from the intersection of the two comparison groups. The pig GAPDH gene was used as the internal control. The primer sequences are shown in Table 1. The purity and concentration of total RNA were measured using a BioPhotometer (Eppendorf, Hamburg, Germany), and all samples showed A260/A280 ratios ranging from 1.93 to 2.05 with RNA concentrations between 182.4 and 205.7 ng/μL, indicating high-quality RNA suitable for downstream analyses. qPCR was performed using the SYBR^®^ Premix Ex Taq™ II kit (Takara, Dalian, China) on an ABI 7500 Real-Time PCR System (Applied Biosystems, Foster City, CA, USA). Each reaction was carried out in a final volume of 15 μL, containing 7.5 μL of 2× SYBR Premix, 1.5 μL of primer mix (each primer used at a final concentration of 2.5 μM), 2.0 μL of cDNA template, and 4.0 μL of RNase-free water. The reaction conditions were as follows: pre-denaturation at 95 °C for 30 s, followed by 45 cycles of 95 °C for 30 s and 60 °C for 34 s, and a final cycle of 95 °C for 30 s, 60 °C for 1 min, and 95 °C for 30 s. Each sample was tested in triplicate to ensure robustness, and relative expression levels were calculated using the 2^−ΔΔCT^ method.

## 3. Results

### 3.1. Sequencing of the Pig Longissimus Dorsi Transcriptome

A sequencing analysis was performed on a library of longissimus dorsi tissues from 12 pigs selected from the CH and HD samples. A total of 12 datasets were obtained, with each dataset yielding approximately 6.4 GB of filtered base pairs (clean bases) after raw data filtering. The obtained data revealed that all the Q20 values were greater than 97%, and all the Q30 values were greater than 93%. The GC content ranged from 49.40% to 52.54%. The results of the quality control analysis indicated that the sequencing results were reliable and suitable for subsequent analysis. See Appendix A for the sequencing results.

### 3.2. Number of Genes in Various FPKM Value Intervals

Inter-sample variations were observed in both the number of expressed genes and their expression levels. The expression values (measured in FPKM) were categorized into distinct intervals to compare expression abundance across samples, as summarized in Table 2. The 0–1 FPKM interval contained the largest proportion of genes, representing approximately 62–67% of all expressed genes. This was followed by the 1–3 and 3–15 FPKM intervals, accounting for approximately 9% and 15%, respectively. In contrast, genes with FPKM values exceeding 60 presented the lowest abundance, accounting for only approximately 2.5% of all expressed genes.

### 3.3. Quantification and Quality Assessment of Gene Expression Levels via RNA-Seq Analysis

In RNA-seq analysis, gene expression levels can be assessed by counting sequencing reads aligned to genomic regions or exonic regions of genes. This process provides a quantitative estimation of gene expression, offering critical insights into gene function and regulatory mechanisms.

As shown in Figure 1a, violin plots and density distribution plots of FPKM (Fragments Per Kilobase of transcript per Million mapped reads) values illustrate the gene expression levels across different samples. Violin plot analysis revealed that the distributions of gene expression levels were largely consistent across groups, with no significant fluctuations, indicating proper sample processing. Similarly, the FPKM density distribution plot (Figure 1b) demonstrated comparable trends among the groups, accurately reflecting the proportion of genes with varying expression levels. Figure 1c provides a visual representation of the intergroup differences and intragroup reproducibility. All four sample groups exhibited high intragroup correlations (R > 0.99), confirming strong reproducibility within replicates. Although the intergroup correlations were slightly lower, they still exceeded 0.9, suggesting robust consistency in gene expression across samples. These results support the validity of differential expression analysis. Furthermore, Figure 1d highlights the high similarity in overall expression distributions among groups, indicating consistent transcriptome profiles across different breeds and sexes. This observation underscores the high sequencing quality and comparability of the datasets. Notably, the median expression levels in the CHF and HDM groups were slightly elevated, possibly due to increased overall expression of certain genes in these populations. Nevertheless, the expression distributions were biologically reasonable without extreme outliers, confirming that sample quality control was satisfactory and that the data are suitable for downstream differential expression analysis.

### 3.4. Differentially Expressed Gene Analysis

To gain deeper insights into the transcriptomic differences in porcine longissimus dorsi muscle, it is essential to identify DEGs between the two breeds within each sex group. In the comparison between CHF and HDF, a total of 5051 DEGs were identified, including 2310 upregulated and 2681 downregulated genes in CHF relative to HDF. Similarly, in the comparison between CHM and HDM, 9972 DEGs were detected, comprising 4984 upregulated and 4988 downregulated genes in CHM compared with HDM. These results are illustrated in Figure 2 and Figure 3. Furthermore, a Venn diagram (Figure 4) was generated to display the shared and unique DEGs between the two comparative groups.

### 3.5. Differential Gene Clustering Analysis

Genes with similar expression patterns often have similar functions or are involved in the same metabolic processes (pathways) [27]. Thus, clustering of genes with similar expression patterns is an analytical strategy that can contribute to the identification of the function of unknown genes or can help to characterize the unknown functions of known genes. To identify clusters with functional enrichment, hierarchical clustering was performed on the basis of gene expression patterns. We performed clustering analysis of the DEGs, and the results are shown in Figure 5. The 12 samples in this study were grouped into four distinct clusters. The CHF group presented a gene expression pattern similar to that of the control HDF group, whereas the CHM group presented relatively low similarity to the control HDM group, indicating notable differences between them. The breed effect was statistically significant, and a certain degree of sex effect was also observed.

### 3.6. GO Enrichment and KEGG Pathway Analyses

To further investigate the differentially expressed genes (DEGs) in the two comparison groups, Gene Ontology (GO) enrichment analysis was performed. The bar plot of GO enrichment for DEGs visually displays the distribution of enriched terms across the three major GO categories: biological process, cellular component, and molecular function. The top 30 most significantly enriched GO terms, ranked by ascending q value, are presented in the bar chart below. If fewer than 30 terms were significantly enriched, all the terms are displayed. The top two significantly enriched GO terms within each category are summarized as follows: in the CHF vs. HDF comparison, biological processes were associated primarily with organonitrogen compound metabolic processes and protein metabolic processes; cellular components, with intracellular and organelle; and molecular functions, with binding and protein binding. In the CHM vs. HDM comparison, biological processes included cellular macromolecule metabolic processes and organonitrogen compound metabolic processes, whereas cellular components and molecular functions were consistently enriched in intracellular, organelle, binding, and protein binding. The results are shown in Figure 6.

Significant GO pathways were identified in both comparison groups using a q value threshold of <0.05. In the CHF vs. HDF comparison, 300 significant biological processes (BP), 53 cellular components (CC), and 64 molecular functions (MF) were obtained. In the CHM vs. HDM comparison group, 725 significant BP, 103 CC, and 115 MF terms were identified. From the significant GO pathways in both comparison groups, GO terms directly related to muscle growth were selected on the basis of their descriptions, and the results are presented in Table 3 and Table 4.

The results of the KEGG enrichment analysis of the DEGs can be visually represented via a scatter plot, as shown in Figure 7. The Rich factor, the Q value, and the number of genes enriched in each pathway were used to evaluate the degree of KEGG enrichment. The Rich factor refers to the ratio of the number of differentially expressed genes enriched in a pathway to the number of annotated genes. A higher Rich factor indicates a greater degree of enrichment. The Q value (range: 0–1) is the *p* value adjusted for multiple hypothesis testing, and a value closer to zero indicates more significant enrichment. Figure 7 displays the top 20 most significantly enriched pathways sorted in ascending order of the Q value. If fewer than 20 pathways were enriched, all are shown. In the CHF vs. HDF and CHM vs. HDM groups, 19 and 13 pathways were significantly enriched (*p* < 0.05), respectively. (Table 5 and Table 6) In the CHF vs. HDF group, the DEGs associated with the focal adhesion, *Staphylococcus aureus* infection, and leishmaniasis pathways were significantly enriched. In the CHM group compared with the HDM group, the differentially expressed genes were significantly enriched in focal adhesion, herpes simplex virus 1 infection, human T-cell leukemia virus 1 infection, and the insulin signalling pathway. Muscle growth primarily involves three pathways, as listed in Table 7.

### 3.7. Protein–Protein Interaction Analysis

Using the cytoNCA plugin in Cytoscape software (version 3.10.3), the protein-protein interaction (PPI) network data of the differentially expressed genes were analysed via the betweenness centrality algorithm. As shown in Figure 8, key genes, including *ITGB1*, *SRC*, *MYL2*, *ITGAV*, *PRKACA*, *MAPK3*, *ACTB*, *ACTG1*, and *PTK2*, were identified in the PPI networks of both comparison groups.

### 3.8. Verification of the Accuracy of the RNA-Seq Data via RT-qPCR

On the basis of the intersecting genes from both comparison groups, eight DEGs were randomly selected for validation of the RNA-seq results via Qpcr. The results revealed that the fold changes in the eight genes in the Qpcr and RNA-seq datasets presented the same trends (Figure 9). The linear regression between the DEG data obtained from the Qpcr and RNA-seq results revealed a high correlation (R^2^ = 0.991), indicating the reliability of our RNA-seq results.

## 4. Discussion

RNA-seq provides a quantitative and open system for large-scale analysis of transcription results [28]. In the animal husbandry industry, meat quality is a critical economic characteristic [29]. Identifying key candidate genes involved in muscle growth is effective in studying molecular genetic regulatory mechanisms for muscle growth and development. These findings are expected to provide a critical scientific basis for developing strategies to enhance muscle development and deliver high-quality meat products. Muscle is an essential part of all meat products. Growth performance and product quality are considered the basic conditions for modern breeding [30]. Muscle growth is regulated by the homeostatic balance of muscle protein biosynthesis and degradation [31,32]. This is the result of the combination of a distinctive genotype, nutrient status, age, and muscle type in muscle tissue [33]. Chinese local pig breeds have universal characteristics of greater lipid deposition capacity and lower lean meat percentages than Western commercial pig breeds do, indicating that these extreme phenotype breeds could influence the variation in pork quality [34]. In the present study, to explore the hereditary elements of meat quality differences, transcriptome profiles of the longissimus dorsi muscle were compared between CH and HD pigs with divergent muscle growth rates.

The sequencing results revealed that the Q20 values for all the samples were greater than 97%, and the Q30 values exceeded 93%, indicating high sequencing quality and providing a solid foundation for subsequent analyses. Notably, the distribution of genes across different expression levels revealed that genes with FPKM values in the 0–1 interval accounted for 62–67% of the total, a pattern consistent with the general characteristics of eukaryotic transcriptomes, where the majority of genes are expressed at low levels or remain silent, while highly expressed genes are relatively scarce [35]. In this study, 5051 differentially expressed genes (DEGs) were identified in the CHF vs. HDF comparison group, and 9972 DEGs were detected in the CHM vs. HDM comparison group. The notably greater number of DEGs in the latter group suggests that transcriptional differences between breeds may be more pronounced in male individuals. This difference may arise from sex-dependent regulatory mechanisms that shape muscle development and transcriptional activity. In male pigs, elevated androgen levels act through the androgen receptor (AR) to modulate downstream anabolic signaling and gene expression, stimulating myogenic differentiation and protein accretion [36,37]. Androgens can also enhance mitochondrial biogenesis and oxidative metabolism by activating co-regulators such as PGC-1α, thereby increasing ATP production to support rapid muscle hypertrophy [38,39]. The testes, as a primary endocrine organ, sustain this anabolic state by maintaining continuous androgen secretion, which contributes to higher metabolic flux, greater amino acid utilization, and elevated transcriptional turnover in muscle tissue. Conversely, estrogen in females promotes a more energy-conserving phenotype, favoring lipid storage over protein accretion and stabilizing metabolic homeostasis [40]. Estrogen also attenuates excessive oxidative stress and suppresses catabolic gene expression, which may contribute to the relatively consistent transcriptional profiles observed in females [41,42]. Collectively, these hormonal and metabolic interactions may partly explain why male individuals exhibit a broader and more dynamic range of transcriptional responses, although further studies are needed to verify the specific molecular mediators underlying this sex-dependent divergence.

Muscle growth is fundamentally dependent on protein synthesis. In the Gene Ontology (GO) enrichment analysis, the differentially expressed genes were predominantly enriched in functional categories such as protein metabolic process and cytoskeletal protein binding. The enrichment results indicated that most significant GO terms were classified under the biological process (BP) category, particularly those related to muscle development and structural remodeling. Within the BP category, key terms primarily associated with skeletal muscle development included muscle contraction, muscle structure development, muscle cell differentiation, cell adhesion, and actin cytoskeleton organization. Genes annotated to muscle contraction regulate sarcomere assembly, actomyosin interaction, and contractile tension, which determine fiber-type composition and contraction efficiency [43,44]. The enrichment of muscle structure development and muscle cell differentiation reflects active regulation of myogenesis, encompassing myoblast proliferation, migration, fusion, and maturation into multinucleated fibers [45,46]. Actin cytoskeleton organization and cell adhesion represent remodeling of the cytoskeletal network and cell–matrix junctions that maintain structural stability and mediate mechanical signaling during hypertrophic growth [47,48]. Cell migration terms showed stronger enrichment in males (CHM vs. HDM), consistent with higher expression of genes involved in cytoskeletal remodeling, focal adhesion, and extracellular matrix interaction. This trend suggests a more dynamic structural turnover potentially modulated by androgen-dependent signaling [36,37]. These enriched processes imply that transcriptional divergence between Liaoning Black and HD involves two major functional modules: (i) structural remodeling, encompassing sarcomere assembly, adhesion, and cytoskeletal reorganization; and (ii) cellular differentiation and migration, facilitating myoblast fusion and fiber regeneration. Such regulatory differences may underlie breed- and sex-specific variation in muscle growth and fiber composition. To further elucidate the molecular mechanisms underlying breed differences, KEGG enrichment analysis identified 19 and 13 significantly enriched pathways in the two comparison groups, respectively. By screening KEGG pathways related to muscle growth, the focal adhesion pathway was found to be significantly enriched in both comparison groups. This pathway has been demonstrated to play a central role in muscle contraction, cell migration, and signal transduction [49]. The focal adhesion pathway plays a central role in converting mechanical stimuli into biochemical signals that regulate muscle cell adhesion, migration, and growth. Integrin receptors, particularly *ITGB1* and *ITGA7*, mediate cell–extracellular matrix (ECM) interactions to anchor myofibers to the basal lamina, thereby maintaining structural integrity and transmitting contractile forces across the sarcolemma [43,50]. Upon integrin engagement, focal adhesion kinase (*FAK*) is recruited and activated, triggering the Src–PI3K–Akt signaling cascade that subsequently stimulates the mTOR pathway to enhance protein synthesis and drive muscle cell growth and hypertrophy [51,52,53]. During myogenesis, FAK–Src signaling coordinates myoblast alignment, migration, and fusion into multinucleated fibers, while in mature muscle it sustains tension-dependent remodeling and structural stability [54,55,56]. The significant enrichment of this pathway in both comparison groups suggests that differences in integrin–cytoskeleton signaling contribute to breed-specific muscle development. CH, characterized by higher intramuscular fat and slower muscle turnover, may rely more on adhesion-mediated structural maintenance, whereas HD exhibit stronger activation of the FAK–mTOR axis associated with rapid hypertrophic growth. Collectively, these results highlight the pivotal role of focal adhesion–related transcriptional regulation in shaping distinct muscle growth patterns among pig breeds. In the CHM vs. HDM comparison group, the insulin signalling pathway was further observed to be significantly enriched. The insulin signaling pathway is a central regulator of skeletal muscle anabolism and metabolism. Activation of the insulin receptor stimulates the IRS–PI3K–Akt cascade, which in turn activates mTORC1 to promote protein synthesis via phosphorylation of *S6K1* and *4E-BP1*, while concurrently inhibiting FoxO-mediated expression of Atrogin-1 and MuRF1 to reduce protein degradation [57]. Within the insulin signaling pathway, Akt-dependent phosphorylation of AS160/TBC1D4 enhances GLUT4 translocation and increases muscle glucose uptake, thereby coupling hormonal and nutrient cues to muscle growth and metabolic efficiency [58,59,60]. Therefore, the insulin signaling pathway plays a dual role in skeletal muscle maintenance: on one hand, it promotes anabolic metabolism, and on the other, it optimizes energy utilization. As insulin signaling is a central regulator of protein synthesis, glucose metabolism, and muscle hypertrophy, its higher enrichment in Duroc males suggests enhanced anabolic and metabolic activity in this lean-type breed. This activation likely supports their rapid muscle deposition and high lean meat ratio. Conversely, CH, characterized by greater fat deposition and slower muscle turnover, may exhibit relatively attenuated anabolic responses through this pathway. Such variation may reflect differences in nutrient partitioning and metabolic strategy between pig breeds rather than absolute differences in insulin pathway expression. Overall, the distinct transcriptional enrichment patterns of the insulin signaling pathway between breeds imply that differential utilization of anabolic signaling contributes to the observed variation in muscle growth rate and meat composition [61].

The hub genes identified in the PPI network—*ITGB1*, *SRC*, *MAPK3*, *PRKACA*, and *MYL2*—constitute a coordinated regulatory axis integrating mechanical signaling, intracellular kinase cascades, and metabolic balance to modulate skeletal muscle development. Collectively, these genes regulate processes such as extracellular matrix adhesion, cytoskeletal remodeling, myogenic differentiation, and contractile specialization, thereby bridging mechanical and metabolic control of muscle growth. To further elucidate the molecular mechanisms by which these key genes participate in muscle growth, each gene was analyzed individually to determine its distinct biological function and regulatory role within this network.

*ITGB1* (integrin β1) acts as a pivotal mechanosensory and signaling molecule in skeletal muscle growth by linking the extracellular matrix (ECM) to the cytoskeleton through focal adhesion complexes. Upon mechanical stimulation such as muscle stretching or contraction, *ITGB1* forms α7β1-integrin heterodimers that activate *FAK* and *SRC*, initiating downstream PI3K–Akt–mTOR and MAPK cascades that promote protein synthesis, sarcomere assembly, and myofiber hypertrophy [62,63,64]. Through the FAK/SRC axis, *ITGB1* also regulates the activation, migration, and fusion of satellite cells by stimulating small GTPases (Rac1, Cdc42) and upregulating myogenic transcription factors such as *MyoD* and *MEF2*, thereby facilitating myogenic differentiation. In addition, ITGB1-mediated activation of the PI3K–Akt–mTOR pathway enhances translational efficiency and ribosome biogenesis, providing a molecular basis for anabolic metabolism and muscle fiber enlargement. Conversely, reduced *ITGB1* expression attenuates focal adhesion signaling and mechanical sensitivity, suppressing anabolic activity and leading to slower oxidative-type muscle growth, as observed in CH. Beyond structural and signaling functions, *ITGB1* also modulates muscle fiber-type composition through crosstalk with the AMPK/mTOR pathway, balancing glycolytic and oxidative metabolism in response to mechanical and metabolic demands [50,65].

The *SRC* (v-src sarcoma viral oncogene homologue) gene encodes a nonreceptor tyrosine kinase that plays a pivotal role in transmitting mechanical and biochemical signals during skeletal muscle development. It acts downstream of the integrin–FAK complex at focal adhesions, where it mediates cytoskeletal remodeling, myoblast migration, and adhesion turnover through phosphorylation of focal adhesion proteins such as paxillin and cortactin [66]. *SRC* also participates in anabolic signaling cascades, including the MAPK/ERK and PI3K–Akt–mTOR pathways, thereby promoting cell proliferation and protein synthesis during myogenic expansion [66]. Conversely, inhibition or downregulation of *SRC* activity enhances p38 MAPK signaling, shifting the cellular state toward differentiation and structural maturation [67]. In this study, the *SRC* gene was downregulated in both comparison groups, suggesting a potential reduction in focal adhesion–mediated signaling activity in the longissimus dorsi muscle of CH. This inference is consistent with the established role of the integrin–FAK–SRC signaling axis in mechanotransduction and skeletal muscle adaptation, where lower *SRC* activity has been shown to reduce cytoskeletal remodeling and anabolic signaling [68]. Moreover, suppression of *SRC* activity can activate the p38 MAPK pathway and promote myogenic differentiation, indicating that the observed downregulation may reflect a transition from proliferative to differentiative muscle states [67]. While this transcriptional pattern may suggest attenuated mechanical signaling and enhanced structural stabilization, direct evidence of such breed-specific regulation in pigs remains limited and warrants further validation.

*MAPK3* (mitogen-activated protein kinase 3) encodes a protein belonging to the MAP kinase family. MAP kinases, also known as extracellular signal-regulated kinases (ERKs), act in a highly conserved signaling cascade that regulates diverse cellular processes—including proliferation, differentiation, and cell cycle progression—in response to extracellular stimuli. ERK1 (*MAPK3*) and *ERK2* (extracellular signal-regulated kinase 2) are major components of the MAPK cascade that transduces growth factor and mechanical signals from the cell membrane to the nucleus, thereby influencing transcriptional programs essential for myogenesis and hypertrophy [69]. Previous studies have demonstrated that *MAPK3* is closely involved in muscle growth and differentiation. For instance, Chiou et al. [70] examined the effects of *MAPK3* in zebrafish and found that its expression is indispensable for embryonic development and cell proliferation, suggesting a conserved role in tissue growth. Similarly, Zhang et al. [71] identified *MAPK3* as a candidate gene associated with muscle development in Inner Mongolian cashmere goats through genome-wide association analysis. Mechanistically, *MAPK3 (ERK1)* functions downstream of growth factors and mechanical cues to activate transcription factors such as *MyoD* and *MEF2*, promoting satellite cell differentiation and myofibril assembly. In addition, ERK1/2 signaling enhances protein synthesis and sarcomere organization by phosphorylating downstream effectors that drive anabolic metabolism and cytoskeletal remodeling. Notably, in this study, *MAPK3* was differentially expressed only in the boar group comparison, consistent with the sex-specific expression reported by Drag et al. [43]. This pattern implies that *MAPK3* activity may be modulated by androgen-dependent pathways, as androgens have been shown to activate ERK signaling via non-genomic mechanisms, thereby promoting anabolic processes and muscle fiber hypertrophy [36,72,73]. Therefore, the observed male-specific expression of MAPK3 may reflect hormone-mediated regulation of the ERK pathway, contributing to sex-dependent differences in muscle growth and transcriptional divergence between pig breeds.

*PRKACA* (Protein Kinase cAMP-Activated Catalytic Subunit Alpha) encodes the catalytic α-subunit of cAMP-dependent protein kinase A (*PKA*), a key serine/threonine kinase mediating cellular responses to hormonal and metabolic stimuli. *PKA* is activated by cAMP generated downstream of β-adrenergic or glucagon receptor stimulation, leading to phosphorylation of target proteins that regulate gene expression, energy metabolism, and protein turnover. In skeletal muscle, the cAMP–PKA pathway orchestrates both anabolic and catabolic processes: *PKA* phosphorylates and activates CREB and mTOR to enhance muscle protein synthesis, while concurrently inhibiting FoxO-mediated ubiquitin–proteasome degradation pathways [74]. Studies in mice have shown that *PRKACA* deficiency causes impaired somatic growth and muscle atrophy, underscoring its essential role in skeletal muscle integrity [75,76]. Moreover, elevated *PKA* activity promotes oxidative slow-twitch fiber specification by regulating calcium-dependent signaling and mitochondrial biogenesis [77,78]. It is noteworthy that *PRKACA* exhibited an upregulated trend in the male comparison group, which may be associated with androgen-induced enhancement of β-adrenergic signaling and subsequent activation of the cAMP–PKA pathway, potentially promoting the phosphorylation of CREB and mTOR, and facilitating muscle protein synthesis and fiber growth. In contrast, *PRKACA* expression was downregulated in the female group, which might reflect estrogen-mediated modulation of energy allocation and lipid metabolism, leading to relatively lower anabolic activity and a tendency toward metabolic homeostasis rather than rapid muscle accretion. Collectively, these observations suggest that the sex-dependent expression pattern of *PRKACA* could influence cAMP–PKA signaling dynamics and thereby contribute to differences in muscle growth and metabolic adaptation between males and females.

*MYL2* (Myosin light chain 2) encodes a regulatory light chain of myosin that plays an essential role in modulating actin–myosin interactions and sarcomere assembly in striated muscle. Its promoter is skeletal muscle–specific and contains multiple binding sites for transcription factors such as *MEF2*, *MyoD*, and *MyoG*, which are key regulators of myoblast proliferation and differentiation [79,80]. *MYL2* phosphorylation influences cross-bridge cycling kinetics and contractile force generation, thereby determining muscle fiber contractility and fiber-type specification through Ca^2+^-dependent signaling and Wnt pathway interaction. In this study, *MYL2* expression was upregulated in both comparison groups of CH, consistent with the findings of Pan et al. [81]. This upregulation suggests enhanced sarcomeric organization and structural remodeling associated with slow oxidative fiber characteristics. Given that CH exhibit higher intramuscular fat content and a greater proportion of type I fibers, the increased *MYL2* expression may reflect a breed-specific adaptation toward improved muscle endurance and meat quality rather than rapid hypertrophic growth. Collectively, these results imply that *MYL2* acts as a structural and regulatory determinant linking transcriptional control with contractile and metabolic specialization in porcine skeletal muscle.

## 5. Conclusions

Taken together, this study provides new insights into the molecular mechanisms governing breed- and sex-specific muscle growth in pigs. By integrating transcriptomic profiling, functional enrichment, and protein–protein interaction network analysis, we identified five hub genes (ITGB1, SRC, MAPK3, PRKACA, and MYL2) that collectively form a regulatory axis linking mechanical signaling, hormonal control, and metabolic adaptation in skeletal muscle. Unlike previous transcriptomic studies that mainly focused on commercial lean breeds, this research is the first to systematically compare an indigenous Chinese breed (Liaoning Black pig) with a Western meat-type breed (Duroc) at the transcriptional level during the finishing phase. The results highlight distinct activation patterns in the focal adhesion and insulin signaling pathways, revealing that HD rely on enhanced integrin–FAK–mTOR signaling for rapid hypertrophy, whereas CH exhibit transcriptional characteristics favoring oxidative metabolism and muscle quality. These findings not only elucidate the genetic and molecular basis of muscle growth divergence between pig breeds but also provide valuable candidate genes and pathways for precision molecular breeding aimed at improving both growth rate and meat quality.

## Figures and Tables

**Figure 1 cimb-47-00917-f001:**
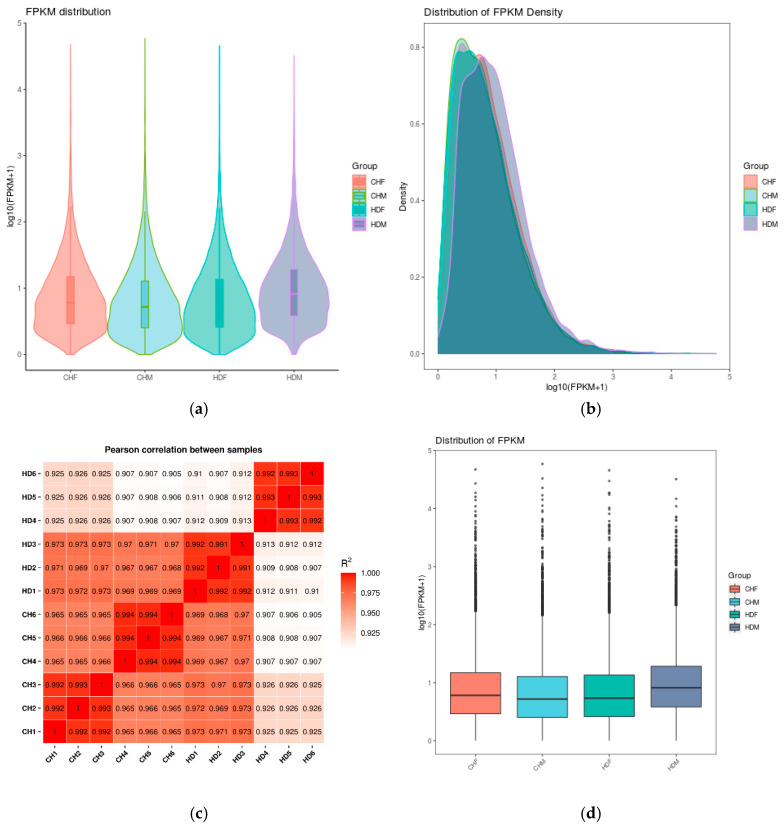
Comparison of gene expression levels under different experimental conditions. Note: (**a**) Violin plot of FPKM values: The *x*-axis represents sample (or group) names, and the *y*-axis represents log10(FPKM + 1). Each violin shape displays five statistics from top to bottom: maximum, upper quartile, median, lower quartile, and minimum. The width of each violin indicates the number of genes at a given expression level. (**b**) Density distribution plot of FPKM: The *x*-axis represents log10(FPKM + 1), and the y-axis represents gene density. (**c**) Heatmap of inter-sample correlation: The numbers in the plot indicate the squared correlation coefficients between samples. (**d**) Box plot of FPKM values: The *x*-axis represents sample (or group) names, and the *y*-axis represents log10(FPKM + 1). Each box displays five statistics from top to bottom: maximum, upper quartile, median, lower quartile, and minimum.

**Figure 2 cimb-47-00917-f002:**
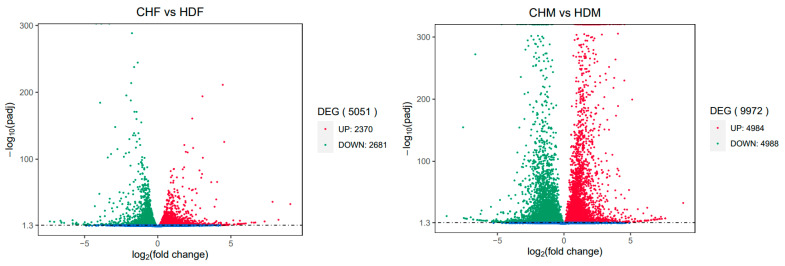
Volcano plot of differentially expressed genes between the two comparison groups. Note: The *x*-axis indicates the fold change in gene expression; the *y*-axis represents the statistical significance of the differential expression. Significantly differentially expressed genes are shown as red dots (upregulated) and green dots (downregulated). Genes without significant differential expression are indicated by blue dots.

**Figure 3 cimb-47-00917-f003:**
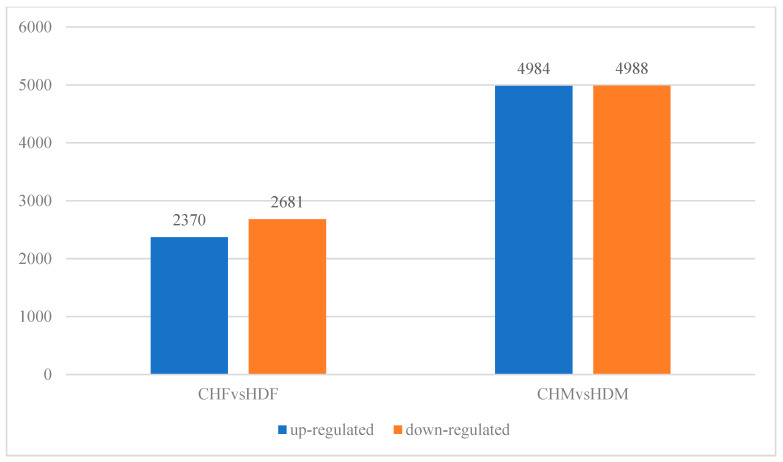
Bar plot of differentially expressed genes between the two comparison groups.

**Figure 4 cimb-47-00917-f004:**
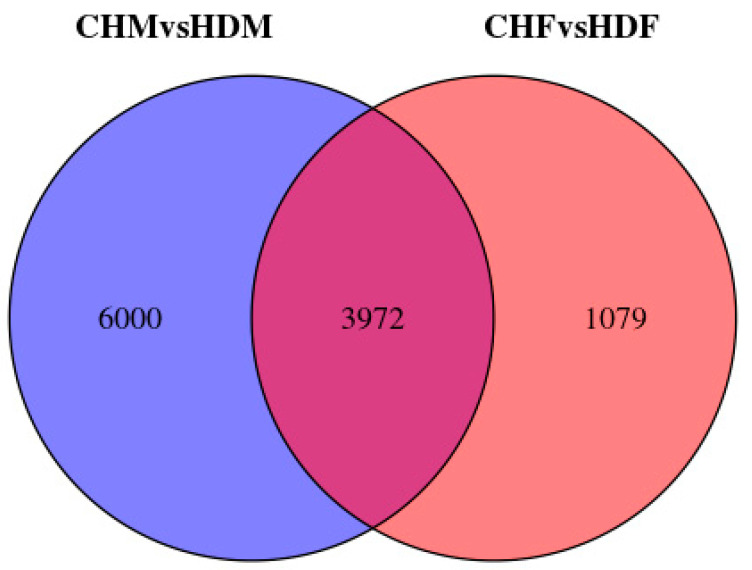
Venn diagram of differentially expressed genes between the two comparison groups.

**Figure 5 cimb-47-00917-f005:**
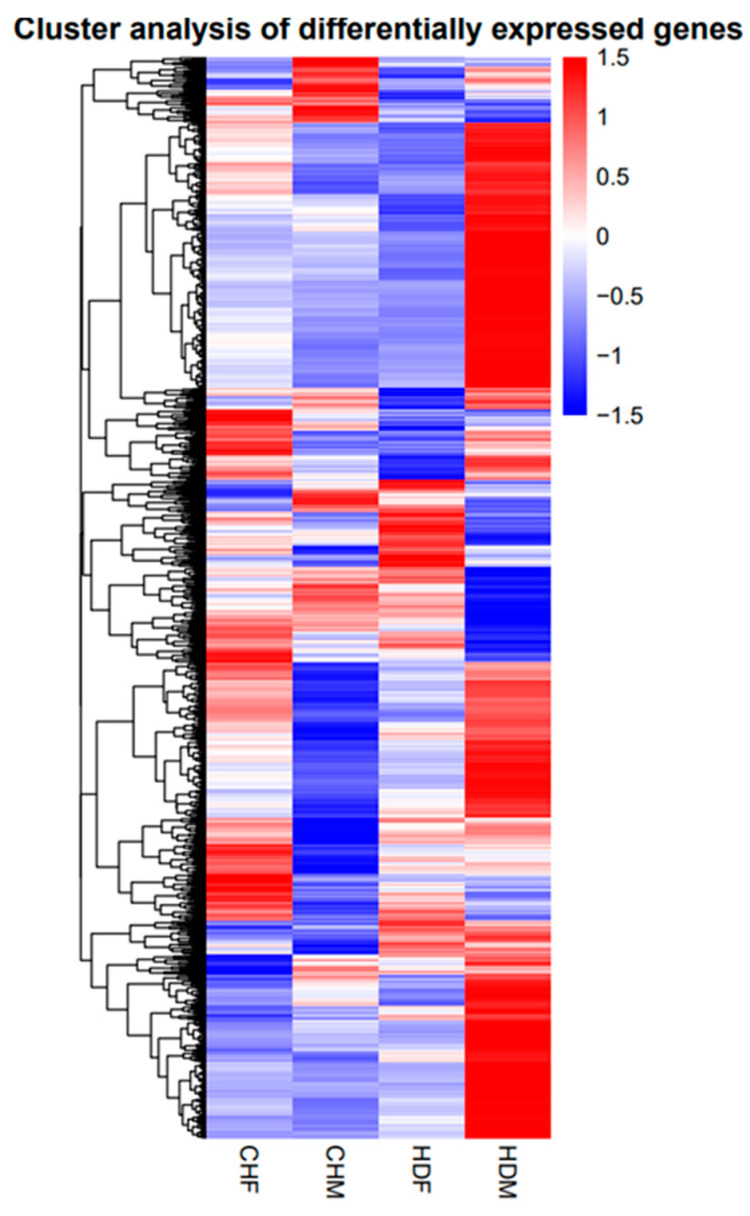
Cluster analysis diagram.

**Figure 6 cimb-47-00917-f006:**
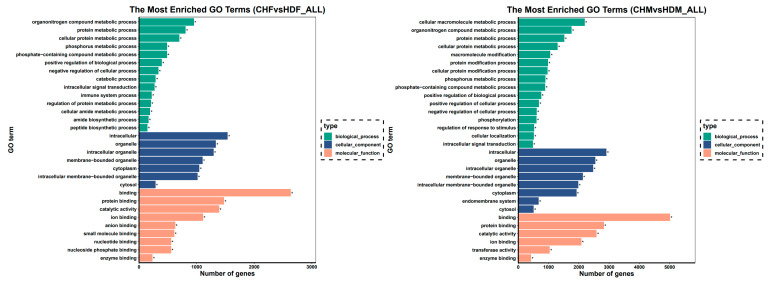
Bar graph of the GO enrichment results. Note: The *y*-axis represents the enriched GO terms, and the *x*-axis indicates the number of differentially expressed genes within each term. Different colours distinguish biological process, cellular component, and molecular function. GO terms marked with an asterisk (*) indicate significant enrichment.

**Figure 7 cimb-47-00917-f007:**
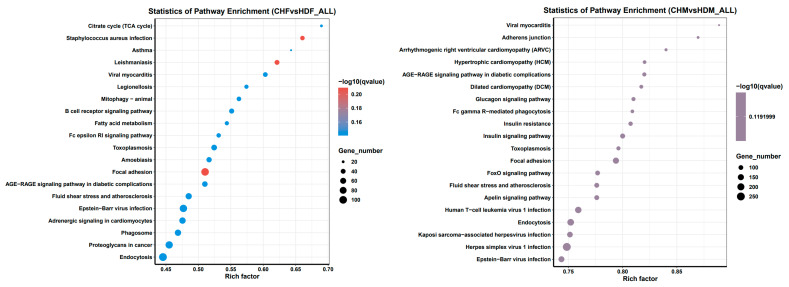
The vertical axis represents pathway names, the horizontal axis represents the Rich factor, the size of the points represents the number of differentially expressed genes in this pathway, and the color of the points corresponds to different Q value ranges.

**Figure 8 cimb-47-00917-f008:**
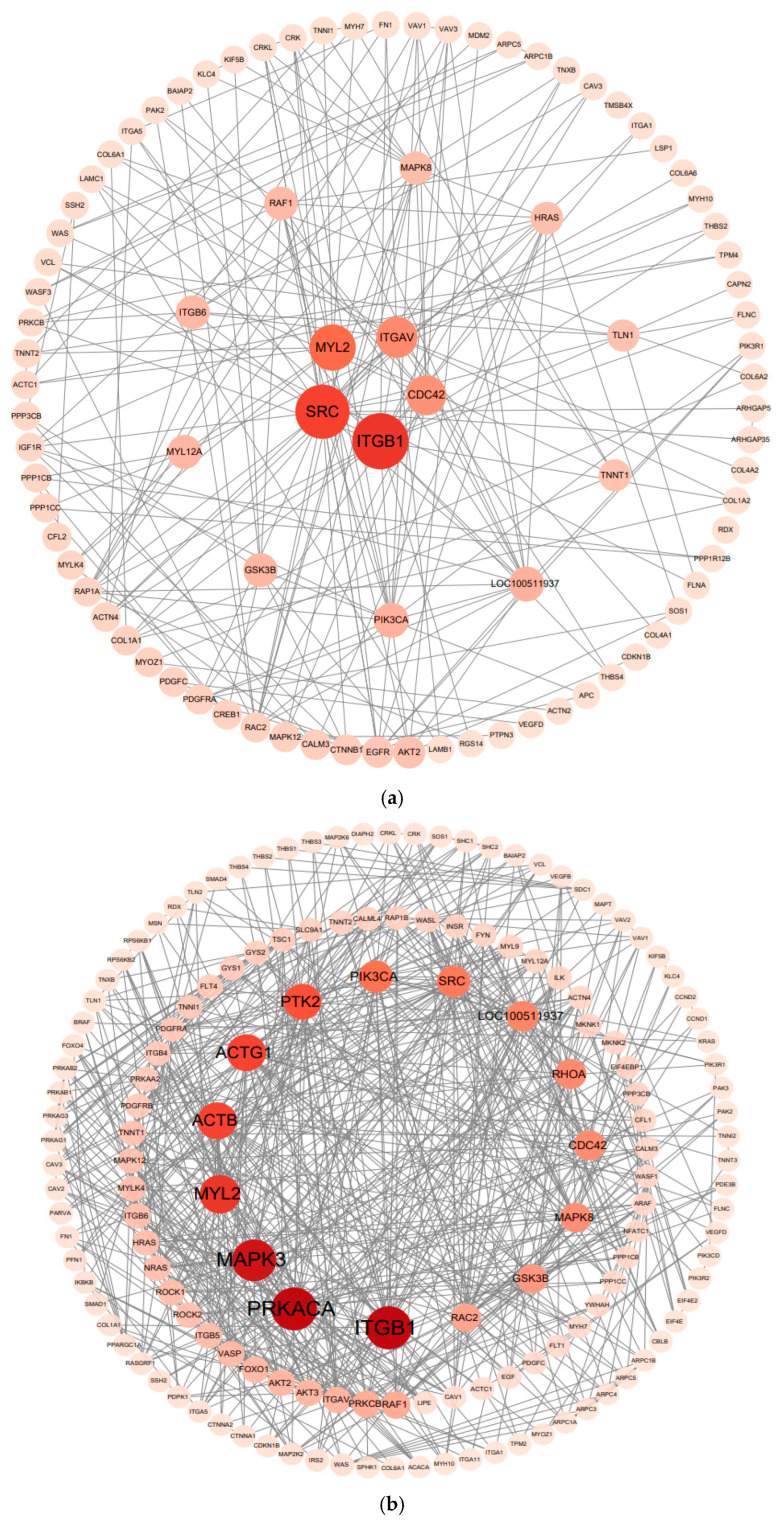
Protein-protein interaction (PPI) network analysis diagram. Protein-protein interaction (PPI) network analysis of the CHF vs. HDF comparison group, (**a**); Protein-protein interaction (PPI) network analysis of the CHM vs. HDM comparison group, (**b**). Note: Color represents the degree of enrichment, with darker colors indicating greater significance; size represents the number of enriched genes, with larger sizes indicating a greater quantity of enriched genes.

**Figure 9 cimb-47-00917-f009:**
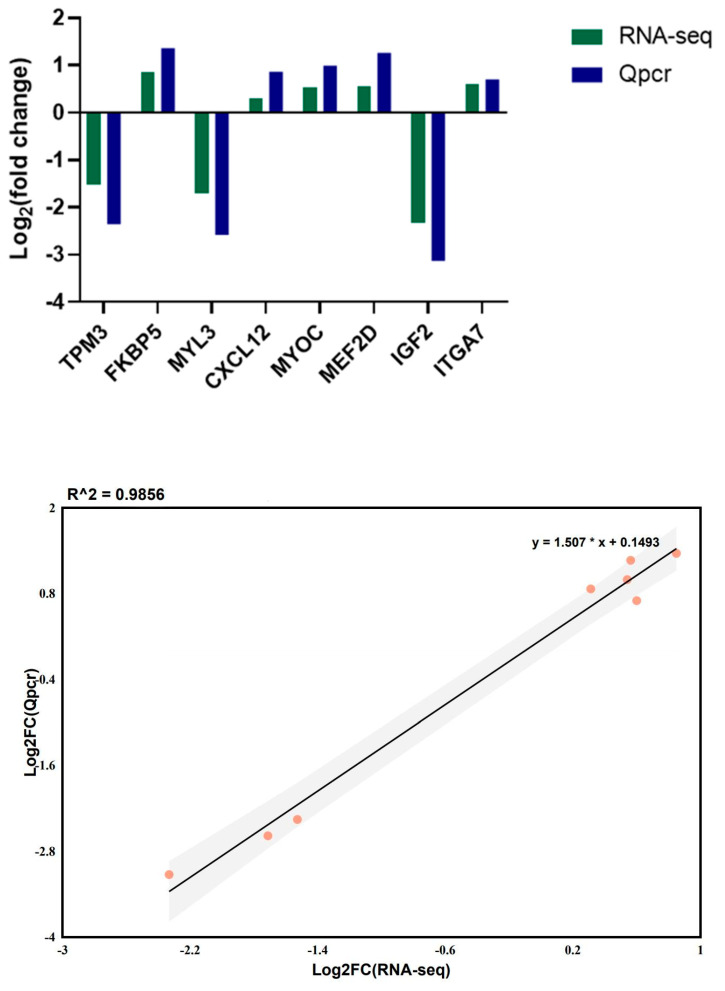
qRT-PCR verification results of DEGs in different pig breeds.

**Table 1 cimb-47-00917-t001:** Primer sequences for reverse transcription–quantitative PCR (RT-qPCR) and PCR.

Gene ID in NCBI	Gene	Primer Sequence (5′→3′)	Product Size (bp)
NM_001001632.1	TPM3-S	GCTGGTTGAAGAGGAGCTGGAC	172
	TPM3-A	TTCTTTGAGTTGGATTTCCTGGAGT	
NM_001315611.2	FKBP5-S	GAACCGTTTGTCTTTAGTCTTGGC	276
	FKBP5-A	CGCTCCTTCGTTGGGATTTG	
NM_001278773.1	MYL3-S	CCAAGGAACCTGAGTTTGATGC	260
	MYL3-A	GAGCATGGGCAGGAACGTGT	
XM_021091233.1	ITGA7-S	TGGTTGGGAGTCAGTGTTCGG	278
	ITGA7-A	CCAAAGAGGAGGTAGTGGCTGT	
NM_001009580.1	CXCL12-S	CACGGCTGAAGAGCAACAATA	101
	CXCL12-A	GAGAGTGGGACTGGGTTTGTTT	
NM_213986.1	MYOC-S	ATGTCTCAGGATGTGCCGTTG	144
	MYOC-A	ACTGCCGGAATGACCAGACAC	
XM_021089672.1	MEF2D-S	GGAGACCTCAACAGTGCTAACG	120
	MEF2D-A	GATGACCTTGTTTAGGCTGTTGC	
NM_213883.2	IGF2(2)-S	GGGCAAGTTCTTCCGCTATGAC	268
	IGF2(2)-A	AGAGGAGGTCACGAGGTGGAT	
NM_001206359.1	gapdh(1)-S	GACATCAAGAAGGTGGTGAAGCA	177
internal control	gapdh(1)-A	GTCGTACCAGGAAATGAGCTTGA	

Note: Upstream primer label: -S (sense primer); downstream primer label: -A (antisense primer).

**Table 2 cimb-47-00917-t002:** Statistics of gene numbers across different expression level intervals.

FPKM Interval	0~1	1~3	3~15	15~60	>60
CH1	20,461 (65.37%)	2963 (9.47%)	5002 (15.98%)	2121 (6.78%)	755 (2.41%)
CH2	20,500 (65.49%)	2996 (9.57%)	4936 (15.77%)	2118 (6.77%)	752 (2.40%)
CH3	20,408 (65.20%)	3026 (9.67%)	4984 (15.92%)	2132 (6.81%)	752 (2.40%)
CH4	21,051 (67.25%)	3023 (9.66%)	4709 (15.04%)	1843 (5.89%)	676 (2.16%)
CH5	21,055 (67.26%)	3013 (9.63%)	4699 (15.01%)	1853 (5.92%)	682 (2.18%)
CH6	21,087 (67.37%)	2986 (9.54%)	4708 (15.04%)	1845 (5.89%)	676 (2.16%)
HD1	20,975 (67.01%)	2958 (9.45%)	4674 (14.93%)	1956 (6.25%)	739 (2.36%)
HD2	20,994 (67.07%)	2915 (9.31%)	4695 (15.00%)	1962 (6.27%)	736 (2.35%)
HD3	21,017 (67.14%)	2913 (9.31%)	4718 (15.07%)	1939 (6.19%)	715 (2.28%)
HD4	19,601 (62.62%)	2711 (8.66%)	5384 (17.20%)	2665 (8.51%)	941 (3.01%)
HD5	19,568 (62.51%)	2710 (8.66%)	5403 (17.26%)	2683 (8.57%)	938 (3.00%)
HD6	19,589 (62.58%)	2705 (8.64%)	5377 (17.18%)	2684 (8.57%)	947 (3.03%)

**Table 3 cimb-47-00917-t003:** CHF vs. HDF comparison group muscle-related GO terms.

GO_Accession	Description	Term_Type	Over-Represented_*p*-Value	Corrected_*p*-Value	DEG_Item
GO:0008092	cytoskeletal protein binding	molecular_function	9.60 × 10^−8^	2.18 × 10^−5^	154
GO:0030016	myofibril	cellular_component	1.11 × 10^−7^	2.40 × 10^−5^	29
GO:0043292	contractile fibre	cellular_component	1.11 × 10^−7^	2.40 × 10^−5^	29
GO:0061061	muscle structure development	biological_process	2.95 × 10^−7^	5.30 × 10^−5^	56
GO:0042692	muscle cell differentiation	biological_process	4.73 × 10^−5^	0.00399	34
GO:0060538	skeletal muscle organ development	biological_process	5.05 × 10^−5^	0.004185	17
GO:0007517	muscle organ development	biological_process	8.59 × 10^−5^	0.006294	30
GO:0007519	skeletal muscle tissue development	biological_process	0.0003	0.017253	15
GO:0060537	muscle tissue development	biological_process	0.000874	0.035947	29
GO:1904706	negative regulation of vascular associated smooth muscle cell proliferation	biological_process	0.001238	0.046245	5

Note: DEG item: Number of differentially expressed genes associated with this GO. The same applies to the table below.

**Table 4 cimb-47-00917-t004:** CHM vs. HDM comparison group muscle-related GO terms.

GO_Accession	Description	Term_Type	Over-Represented_*p*-Value	Corrected_*p*-Value	DEG_Item
GO:0003012	muscle system process	biological_process	2.65 × 10^−8^	1.84 × 10^−6^	56
GO:0061061	muscle structure development	biological_process	1.20 × 10^−7^	6.93 × 10^−6^	85
GO:0090257	regulation of muscle system process	biological_process	6.81 × 10^−7^	3.42 × 10^−5^	39
GO:0006936	muscle contraction	biological_process	9.41 × 10^−7^	4.60 × 10^−5^	48
GO:0006937	regulation of muscle contraction	biological_process	2.19 × 10^−5^	0.000774	32
GO:0006941	striated muscle contraction	biological_process	2.37 × 10^−5^	0.000828	27
GO:0042692	muscle cell differentiation	biological_process	3.27 × 10^−5^	0.001084	51
GO:0007517	muscle organ development	biological_process	0.000198	0.005153	44
GO:0060048	cardiac muscle contraction	biological_process	0.000454	0.010173	20
GO:0006942	regulation of striated muscle contraction	biological_process	0.000933	0.018673	21
GO:0055117	regulation of cardiac muscle contraction	biological_process	0.002365	0.040228	17
GO:1990874	vascular associated smooth muscle cell proliferation	biological_process	0.002622	0.043526	9
GO:0005865	striated muscle thin filament	cellular_component	0.002957	0.048141	8
GO:0030016	myofibril	cellular_component	2.72 × 10^−8^	1.86 × 10^−6^	41
GO:0008092	cytoskeletal protein binding	molecular_function	4.50 × 10^−15^	1.15 × 10^−12^	277

**Table 5 cimb-47-00917-t005:** CHF vs. HDF: Significantly Enriched KEGG Pathways.

Term	Database	ID	Input Number	*p*-Value	Up	Down
Focal adhesion	KEGG PATHWAY	ssc04510	99	0.004337	44	55
Staphylococcus aureus infection	KEGG PATHWAY	ssc05150	35	0.005195	34	1
Leishmaniasis	KEGG PATHWAY	ssc05140	41	0.00574	32	9
Viral myocarditis	KEGG PATHWAY	ssc05416	38	0.010645	27	11
Epstein–Barr virus infection	KEGG PATHWAY	ssc05169	93	0.020259	49	44
B cell receptor signaling pathway	KEGG PATHWAY	ssc04662	43	0.02028	23	20
Toxoplasmosis	KEGG PATHWAY	ssc05145	54	0.020284	32	22
Citrate cycle (TCA cycle)	KEGG PATHWAY	ssc00020	20	0.022353	2	18
Mitophagy-animal	KEGG PATHWAY	ssc04137	36	0.025916	14	22
Legionellosis	KEGG PATHWAY	ssc05134	31	0.030725	15	16
AGE-RAGE signaling pathway in diabetic complications	KEGG PATHWAY	ssc04933	51	0.032827	28	23
Amoebiasis	KEGG PATHWAY	ssc05146	47	0.034029	31	16
Fluid shear stress and atherosclerosis	KEGG PATHWAY	ssc05418	65	0.036342	26	39
Adrenergic signaling in cardiomyocytes	KEGG PATHWAY	ssc04261	68	0.043249	23	45
Asthma	KEGG PATHWAY	ssc05310	18	0.044906	18	0
Proteoglycans in cancer	KEGG PATHWAY	ssc05205	91	0.046646	49	42
Fatty acid metabolism	KEGG PATHWAY	ssc01212	31	0.048777	11	20
Endocytosis	KEGG PATHWAY	ssc04144	106	0.049658	54	52
Fc epsilon RI signaling pathway	KEGG PATHWAY	ssc04664	34	0.049918	18	16

Note: Input number: the number of differentially expressed genes associated with this pathway; up: the number of differentially expressed genes upregulated in this pathway; down: the number of differentially expressed genes downregulated in this pathway. The same applies to the table below.

**Table 6 cimb-47-00917-t006:** CHM vs. HDM: Significantly Enriched KEGG Pathways.

Term	Database	ID	Input Number	*p*-Value	Up	Down
Focal adhesion	KEGG PATHWAY	ssc04510	154	0.014182	54	100
Herpes simplex virus 1 infection	KEGG PATHWAY	ssc05168	253	0.014817	100	153
Human T-cell leukemia virus 1 infection	KEGG PATHWAY	ssc05166	167	0.030231	74	93
Insulin signaling pathway	KEGG PATHWAY	ssc04910	108	0.030592	73	35
Endocytosis	KEGG PATHWAY	ssc04144	179	0.031543	89	90
Viral myocarditis	KEGG PATHWAY	ssc05416	56	0.032809	19	37
Adherens junction	KEGG PATHWAY	ssc04520	60	0.036048	25	35
AGE-RAGE signaling pathway in diabetic complications	KEGG PATHWAY	ssc04933	82	0.038482	33	49
Insulin resistance	KEGG PATHWAY	ssc04931	88	0.041252	55	33
Glucagon signaling pathway	KEGG PATHWAY	ssc04922	81	0.046426	55	26
Dilated cardiomyopathy (DCM)	KEGG PATHWAY	ssc05414	76	0.046815	38	38
Arrhythmogenic right ventricular cardiomyopathy (ARVC)	KEGG PATHWAY	ssc05412	63	0.048268	30	33
Hypertrophic cardiomyopathy (HCM)	KEGG PATHWAY	ssc05410	73	0.048275	35	38

**Table 7 cimb-47-00917-t007:** KEGG pathways related to muscle growth.

	Term	Database	ID	Input Number	*p*-Value	Up	Down
CHFvsHDF	Focal adhesion	KEGG PATHWAY	ssc04510	99	0.004337	44	55
CHMvsHDM	Focal adhesion	KEGG PATHWAY	ssc04510	154	0.014182	54	100
	Insulin signalling pathway	KEGG PATHWAY	ssc04910	108	0.030592	73	35

## Data Availability

The original contributions presented in this study are included in the article. Further inquiries can be directed to the corresponding author.

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
