# Peer review of "Identification of Key Candidate Genes for Muscle Growth in Liaoning Black Pigs and Duroc Pigs via Longissimus Dorsi Muscle Transcriptome Analysis"

_cimb, 2025, doi:10.3390/cimb47110917_

Round 1
Reviewer 1 Report
Comments and Suggestions for Authors
The submitted manuscript addresses a very interesting topic that may help reveal relationships with pig growth characteristics and, consequently, improve breeding methods.
However, several revisions should be made in the manuscript to improve and clarify the overall presentation of the document.
For better clarity, all Latin names and gene names should be written in italics.
The introduction should discuss the genetic aspects related to meat performance in more detail, as this is the main focus of the manuscript.
In the methods section, several details need clarification—for example, could the difference in animal age (26 days) have impacted the results?
Where and how was the muscle sampled (biopsy in live animals or post-mortem)?
Within what time frame was the RNA isolated from tissue and how were the samples stored until then?
Why was only one reference gene selected for comparison?
For clarity, it might be better to move Table 3 to the supplementary material.
The discussion is very extensive; however, I recommend emphasizing the main contributions arising from this study.
Reviewer 2 Report
Comments and Suggestions for Authors
This study compared the transcriptome differences of longissimus dorsi muscle between Liaoning Black pigs and Duroc pigs through RNA sequencing, revealing significant disparities in gene expression between the two breeds, especially in male individuals. The research identified the insulin signaling pathway and focal adhesion pathway as key networks regulating muscle development and screened critical molecular targets such as ITGB1 and SRC. The results provide a theoretical foundation and precise regulatory targets for molecular design breeding in pigs, which is of significant importance for enhancing meat production efficiency and ensuring food safety. However, this manuscript has the following multiple deficiencies that need to be revised.
1 The introduction section is relatively brief. It is suggested to add content from the following three aspects: 1) The history of breed formation and genetic background; 2) The regulatory window for the conversion of muscle fiber types; 3) The landscape of transcriptome research gaps.
2 The healthy CHs (three males and three females) and HDs (three males and three females) selected for this study independently breed individuals aged 6-7 months with comparable body weights. As shown in Table 1.
1)Table 1 seems to be of little significance, and the content can be fully described in the form of text. 2)Please provide a detailed introduction to these two breeds of pigs. What are the weights of the two breeds of pigs at 228 days and 202 days, respectively? What is the significance of choosing these two time points? 3)Please supplement the growth environment of these two breeds of pigs, such as temperature, humidity, and light management conditions.
3 Total RNA was extracted from the longissimus lumborum muscle using TRIzol® Reagent...
Please provide the RNA Integrity Number (RIN) or specific quality assessment criteria.
4 DEGs were identified using the criterion of a q value <0.05 (Benjamini–Hochberg adjusted p value following multiple testing correction).
What is the threshold for fold change in expression levels?
5 Gene ontology enrichment analysis of differentially expressed genes was implemented via the GOseq R package[15], in which gene length bias was corrected.
Is the background gene set used in the GO enrichment analysis the set of all expressed genes?
6 The protein–protein interaction network of the DEGs was constructed via STRING[17] (v12.0)...
Please specify the confidence threshold or species settings used in the STRING database.
7 QPCR was performed via an SYBR® Premix Ex Taq™ II kit (Takara, China) on an ABI 7500 Real-Time PCR System...
Please provide details such as the total volume of the qPCR reaction system and the primer concentration.
8 Please check the entire text to confirm whether it is CH or LH, and use a consistent expression.
9 In the comparison between CHF patients and HDFs, a total of 5,051 DEGs were identified...
It is patients???
10 In this study, 5,051 differentially expressed genes (DEGs) were identified in the CHF vs. HDF comparison group, and 9,972 DEGs were detected in the CHM vs. HDM comparison group.
Please conduct a functional classification of DEGs (such as metabolic pathways, signaling pathways, etc.) and discuss the potential roles of these genes in muscle growth.
11 The notably greater number of DEGs in the latter group suggests that transcriptional differences between breeds may be more pronounced in male individuals.
The text mentions that there are more DEGs in male individuals; please delve into the biological mechanisms underlying this sex difference.
12 Through screening GO categories related to muscle growth in the GO terms, many DEGs were obtained, revealing that the DEGs in both comparison groups were enriched primarily in the BP category.
Please conduct a detailed analysis of the key GO terms in the BP category, such as “muscle contraction” and “cell migration,” and discuss the roles of these processes in muscle growth.
13 By screening KEGG pathways related to muscle growth, the focal adhesion pathway was found to be significantly enriched in both comparison groups.
Please provide a detailed discussion on the role of the focal adhesion pathway in muscle cell adhesion, migration, and signal transduction.
14 In the CHM vs. HDM comparison group, the insulin signaling pathway was further observed to be significantly enriched.
Please provide a detailed discussion on how the insulin signaling pathway promotes protein synthesis and inhibits protein degradation through the PI3K/Akt/mTOR pathway.
15 Protein-protein interaction (PPI) network analysis of differentially expressed genes revealed several key hub genes closely associated with muscle development and signal transduction.
Please provide a detailed discussion of the mechanisms by which these key genes are involved in muscle growth.
16 While this study provides insights into the transcriptomic differences underlying muscle growth, several limitations should be considered.
Please provide a detailed discussion on the impact of these limitations on the results and suggest directions for future research.
17 Please modify the format of the references according to the requirements of the magazine.
Round 2
Reviewer 2 Report
Comments and Suggestions for Authors
The author has already answered or modified the doubts or questions I raised.